# Adversarial robustness assessment: Why in evaluation both $L_0$ and $L_\infty$ attacks are necessary

**Shashank Kotyan** [1]*, **Danilo Vasconcellos Vargas** [1,2]

**1** Department of Information Science and Engineering, Kyushu University, Fukuoka, Japan, **2** Department of Electrical Engineering and Information Systems, School of Engineering, The University of Tokyo, Tokyo, Japan

* shashankkotyan@gmail.com

**Data Availability Statement:** All the code and data is available in the Public Github Repository "https://github.com/shashankkotyan/DualQualityAssessment." Our study uses the public

## Abstract

There are different types of adversarial attacks and defences for machine learning algorithms which makes assessing the robustness of an algorithm a daunting task. Moreover, there is an intrinsic bias in these adversarial attacks and defences to make matters worse. Here, we organise the problems faced: a) Model Dependence, b) Insufficient Evaluation, c) False Adversarial Samples, and d) Perturbation Dependent Results. Based on this, we propose a model agnostic adversarial robustness assessment method based on $L_0$ and $L_\infty$ distance-based norms and the concept of robustness levels to tackle the problems. We validate our robustness assessment on several neural network architectures (WideResNet, ResNet, AllConv, DenseNet, NIN, LeNet and CapsNet) and adversarial defences for image classification problem. The proposed robustness assessment reveals that the robustness may vary significantly depending on the metric used (i.e., $L_0$ or $L_\infty$). Hence, the duality should be taken into account for a correct evaluation. Moreover, a mathematical derivation and a counter-example suggest that $L_1$ and $L_2$ metrics alone are not sufficient to avoid spurious adversarial samples. Interestingly, the threshold attack of the proposed assessment is a novel $L_\infty$ black-box adversarial method which requires even more minor perturbation than the One-Pixel Attack (only 12% of One-Pixel Attack's amount of perturbation) to achieve similar results. We further show that all current networks and defences are vulnerable at all levels of robustness, suggesting that current networks and defences are only effective against a few attacks keeping the models vulnerable to different types of attacks.

## Introduction

Neural networks have empowered us to obtain high accuracy in several natural language processing and computer vision tasks. Most of these tasks are only feasible with the aid of neural networks. Despite these accomplishments, neural networks have been shown to misclassify if small perturbations are added to original samples. Further, these perturbed adversarial samples suggest that conventional neural network architectures cannot understand concepts or high-level abstractions, as we earlier speculated.

dataset CIFAR-10 available from "https://www.cs.toronto.edu/∼kriz/cifar.html".

**Funding:** DV, JST-Act-I Grant Number JP-50243, https://www.jst.go.jp/kisoken/act-i/index.html. DV, JSPS KAKENHI Grant Number JP20241216, https://www.jsps.go.jp/english/e-grants/.

**Competing interests:** The authors have declared that no competing interests exist.

Security and safety risks created by these adversarial samples also prohibit the use of neural networks in many critical applications such as autonomous vehicles. Therefore, it is of utmost significance to formulate not only accurate but robust neural networks. Thus, a robustness assessment is required to evaluate robustness efficiently without in-depth knowledge of adversarial machine learning.

The field of adversarial machine learning has contributed to developing tools that could help develop assessment for robustness. However, the sheer amount of scenarios: attacking methods, defences and metrics ($L_0$, $L_1$, $L_2$ and $L_\infty$) make the current state-of-the-art difficult to perceive. Moreover, most of the recent adversarial attacks and defences are white-box which can not be used in general to assess hybrids, non-standard neural networks and other classifiers.

It turns out that a simple robustness assessment is a daunting task, given the vast amount of possibilities and definitions along with their exceptions and trade-offs. Further, these adversarial samples point out shortcomings in the reasoning of current machine learning algorithms. Improvements in robustness should also result in learning systems that can better reason over data and achieve a new level of abstraction. A robustness assessment would thus help check failures for both reasoning and high-level abstractions.

We formalise some of the problems which must be tackled to create a robustness assessment procedure;

## P1 model dependence

A model agnostic robustness assessment is crucial to compare neural networks with other approaches that may be completely different, like logic hybrids, evolutionary hybrids, and others.

## P2 insufficient evaluation

There are several potential types of adversarial attack variations and scenarios, each with its own bias. The attacks differ substantially depending on metrics optimized, namely $L_0$, $L_1$, $L_2$ and $L_\infty$. However, not all of them are vital for the evaluation of robustness. A robustness assessment should have few but sufficient tests to provide an in-depth analysis without compromising its utility.

## P3 false adversarial samples

Adversarial attacks sometimes produce misleading adversarial samples (samples that can not be recognised even by a human observer). Such deceptive adversarial samples can only be detected through inspection, which causes the evaluation to be error-prone. There is a need for inspection in a robustness assessment and the non-feasibility of fraudulent adversarial samples.

## P4 perturbation dependent results

A varying amount of perturbation leads to varying adversarial accuracy. Moreover, networks differ in their sensitivity to attacks, given a varied amount of perturbation. Consequently, this might result in double standards or hide important information.

Table 1 summarises the existence of the problems in the existing literature.

## Contributions

In this article, we propose a robustness assessment to tackle the problems mentioned above with the following features:

**Table 1. Problems in robustness assessment of other literatures.**

| Literature | P1 | P2 | P3 | P4 |
|---|:---:|:---:|:---:|:---:|
| Chen et al. (2020) [1] | ✓ | ✓ | ✓ | ✓ |
| Croce and Hein (2020) [2] | ✓ | ✓ | | |
| Ghiasi et al. (2020) [3] | ✓ | ✓ | | |
| Hirano and Takemoto (2020) [4] | ✓ | ✓ | ✓ | |
| Cohen et al. (2019) [5] | ✓ | ✓ | | ✓ |
| Tan and Shokri (2019) [6] | ✓ | ✓ | ✓ | ✓ |
| Wang et al. (2019) [7] | ✓ | ✓ | | ✓ |
| Wong et al. (2019) [8] | ✓ | ✓ | | |
| Zhang et al. (2019a) [9] | ✓ | ✓ | | |
| Zhang et al. (2019b) [10] | ✓ | ✓ | | |
| Brendel et al. (2018) [11] | | ✓ | ✓ | ✓ |
| Buckman et al. (2018) [12] | | ✓ | | ✓ |
| Gowal et al. (2018) [13] | ✓ | ✓ | | |
| Grosse et al. (2018) [14] | | ✓ | ✓ | ✓ |
| Guo et al. (2018) [15] | | ✓ | | ✓ |
| Madry et al. (2018) [16] | ✓ | ✓ | | ✓ |
| Singh et al. (2018) [17] | ✓ | ✓ | | ✓ |
| Song et al. (2018) [18] | | ✓ | | |
| Tramer et al. (2018) [19] | ✓ | ✓ | | |
| Arpit et al. (2017) [20] | ✓ | ✓ | | |
| Carlini and Wagner (2017) [21] | ✓ | | | ✓ |
| Chen et al. (2017a) [22] | ✓ | ✓ | | |
| Chen et al. (2017b) [23] | ✓ | ✓ | ✓ | ✓ |
| Das et al. (2017) [24] | | ✓ | | ✓ |
| Gu et al. (2017) [25] | ✓ | | ✓ | ✓ |
| Jang et al. (2017) [26] | ✓ | ✓ | ✓ | ✓ |
| Moosavi et al. (2017) [27] | ✓ | ✓ | | ✓ |
| Xu et al. (2017) [28] | | | | ✓ |
| Kurakin et al. (2016) [29] | ✓ | ✓ | | |
| Moosavi et al. (2016) [30] | ✓ | ✓ | ✓ | ✓ |
| Papernot et al. (2016a) [31] | ✓ | ✓ | | |
| Papernot et al. (2016b) [32] | ✓ | ✓ | | ✓ |
| Goodfellow et al. (2014) [33] | ✓ | ✓ | | |

## Non-gradient based black-box attack (Address P1)

Black-box attacks are desirable for a model agnostic evaluation that does not depend on specific features of the learning process, such as gradients. Our proposed robustness assessment is based on black-box attacks, an existing $L_0$ black-box attack, and a novel $L_\infty$ black-box attack.

## Dual evaluation (Address P2 and P3)

We propose to use attacks solely based on $L_0$ and $L_\infty$ norm to avoid creating adversarial samples which are not correctly classified by human beings after modification. These metrics impose a constraint over the spatial distribution of noise which guarantees the quality of the adversarial sample. This is explained in detail mathematically and illustrated with a counterexample in a later section.

## Robustness levels (Address P4)

We define robustness levels in terms of the constraint's threshold *th*. We then compare multiple robustness levels of results with their respective values at the same robustness level. Robustness levels constrain the comparison of equal perturbation, avoiding comparing results with different degrees of perturbation. Robustness levels add a concept that may aid in the classification of algorithms. For example, an algorithm that is robust to 1-Pixel Attack belongs to the 1-pixel-safe category.

## Related works

It was shown in [34] that neural networks behave oddly for almost the same images. Afterwards, in [35], the authors demonstrated that neural networks show high confidence when presented with textures and random noise. This led to discovering a series of vulnerabilities in neural networks, which were then exploited by adversarial attacks.

Universal adversarial perturbation was shown to be possible in [27], which can be added to most of the samples to fool a neural network. The addition of patches in an image is also shown to make neural misclassify [36]. Moreover, an extreme attack was shown to be effective in which it is possible to make neural networks misclassify with a single-pixel change [37].

Many of these attacks can easily be made into real-world threats by printing out adversarial samples, as shown in [29]. Moreover, carefully crafted glasses can also be made into attacks [38]. Alternatively, even general 3D adversarial objects were shown possible [39].

Regarding understanding the phenomenon, it is argued in [33] that neural networks' linearity is one of the main reasons. Another investigation proposes the conflicting saliency added by adversarial samples as the reason for misclassification [40]. A geometric perspective is analysed in [41], where it is shown that adversarial samples lie in shared subspace, along which the decision boundary of a classifier is positively curved. Further, in [42], a relationship between sensitivity to additive perturbations of the inputs and the curvature of the decision boundary of deep networks is shown. Another aspect of robustness is discussed in [16], where authors suggest that the capacity of the neural networks' architecture is relevant to the robustness. It is also stated in [43] that the adversarial vulnerability is a significant consequence of the dominant supervised learning paradigm and a classifier's sensitivity to well-generalising features in the known input distribution. Also, research by [44] argues that adversarial attacks are entangled with the interpretability of neural networks as results on adversarial samples can hardly be explained.

Many defensive and detection systems have also been proposed to mitigate some of the problems. However, there are still no current solutions or promising ones which can negate the adversarial attacks *consistently*. Regarding defensive systems, there are many variations of defenses [12, 15, 18, 24, 28, 45–47] which are carefully analysed in [48, 49] and many of their shortcomings are documented.

Defensive distillation, in which a smaller neural network squeezes the content learned by the original one, was proposed as a defence [32]. However, it was shown not to be robust enough in [21]. Adversarial training was also proposed, in which adversarial samples are used to augment the training dataset [16, 33, 50]. Augmentation of the dataset is done so that the neural networks can classify the adversarial samples, thus increasing their robustness. Although adversarial training can increase the robustness slightly, the resulting neural network is still vulnerable to attacks [19].

Regarding detection systems, a study from [51] demonstrated that indeed some adversarial samples have different statistical properties which could be exploited for detection. The authors in [28] proposed to compare the prediction of a classifier with the prediction of the

same input, but 'squeezed'. This technique allowed classifiers to detect adversarial samples with small perturbations. Many detection systems fail when adversarial samples deviate from test conditions [52–54]. Thus, the clear benefits of detection systems remain inconclusive.

## Adversarial machine learning as optimisation problem

Let us suppose that for the image classification problem, $x \in \mathbb{R}^{m \times n \times c}$ be the benign image that is to be classified. Here $m$, $n$ is the image's width and height, and $c$ is the number of colour channels. A neural network is composed of several neural layers linked together. Each neural layer is composed of a set of perceptrons (artificial neurons). Each perceptron maps a set of inputs to output values with an activation function. Thus, function of the neural network (formed by a chain) can be defined as:

$$F(x) = f^{(k)}(\ldots f^{(2)}(f^{(1)}(x)))$$ (1)

where $f^{(i)}$ is the function of the $i^{\text{th}}$ layer of the network, where $i = 1, 2, 3, \ldots, k$ and $k$ is the last layer of the neural network. In the image classification problem, $F(x) \in \mathbb{R}^N$ is the probabilities (confidence) for all the available $N$ classes. Hence, argmax($F(x)$) gives the highest confidence class, which is termed as classification of $x$ by the neural network $F$.

Before formally defining the problem of adversarial machine learning, let us also define adversarial samples $\hat{x}$ as:

$$\hat{x} = x + \epsilon_x$$ (2)

in which $\epsilon_x \in \mathbb{R}^{m \times n \times c}$ is the perturbation added to the input. Further by the definition of adversarial samples that the highest confidence class of original image ($x$) and adversarial sample ($\hat{x}$) should differ, we can impose a constraint on $\epsilon_x$ such as,

$$\{\hat{x} \in \mathbb{R}^{m \times n \times 3} \mid \underset{C}{\text{argmax}}(F(x)) \neq \underset{\hat{C}}{\text{argmax}}(F(\hat{x}))\}$$ (3)

Here, $C$ and $\hat{C}$ are the respective classfications for original image ($x$) and adversarial sample ($\hat{x}$) by the same network $F$.

Based on the above definition, adversarial machine learning can be seen as a constrained optimisation problem. The constraint has the objective of disallowing perturbations that could make $x$ unrecognisable or change its correct class. Therefore, the constraint is itself a mathematical definition of what constitutes an imperceptible perturbation. Many different norms are used in the literature (e.g., $L_0$, $L_1$, $L_2$, and $L_\infty$). Intuitively, the norms allow for different types of attacks. $L_0$ allows attacks to perturb a few pixels strongly, $L_\infty$ allow all pixels to change slightly, and both $L_1$ and $L_2$ allow for a mix of both strategies.

Making use of the definition of adversarial samples, adversarial machine learning thus, can be formally defined as the following optimization problem for untargeted black-box attacks:

$$\underset{\epsilon_x}{\text{minimize}} \qquad F(x + \epsilon_x)_C$$
$$\text{subject to} \qquad \|\epsilon_x\|_p \leq th$$ (4)

Similarly optimization problem for the targeted black-box attacks can be defined as:

$$\underset{\epsilon_x}{\text{maximize}} \qquad F(x + \epsilon_x)_T$$
$$\text{subject to} \qquad \|\epsilon_x\|_p \leq th$$ (5)

**Table 2. Norms used in robustness assessment in other literatures.**

| Literature | $L_0$ | $L_1$ | $L_2$ | $L_\infty$ |
|---|---|---|---|---|
| Chen et al. (2020) [1] | | | ✓ | ✓ |
| Croce and Hein (2020) [2] | | | ✓ | ✓ |
| Ghiasi et al. (2020) [3] | | | ✓ | ✓ |
| Hirano and Takemoto (2020) [4] | | | ✓ | |
| Cohen et al. (2019) [5] | | | ✓ | |
| Tan and Shokri (2019) [6] | ✓ | | | |
| Wang et al. (2019) [7] | | | ✓ | |
| Wong et al. (2019) [8] | | | | ✓ |
| Zhang et al. (2019a) [9] | | | | ✓ |
| Zhang et al. (2019b) [10] | | | | ✓ |
| Brendel et al. (2018) [11] | | | ✓ | |
| Buckman et al. (2018) [12] | | | | ✓ |
| Gowal et al. (2018) [13] | | | | ✓ |
| Grosse et al. (2018) [14] | | | ✓ | |
| Guo et al. (2018) [15] | | | ✓ | ✓ |
| Madry et al. (2018) [16] | | | ✓ | ✓ |
| Singh et al. (2018) [17] | | | | ✓ |
| Song et al. (2018) [18] | | | | ✓ |
| Tramer et al. (2018) [19] | | | | ✓ |
| Arpit et al. (2017) [20] | | | | ✓ |
| Carlini and Wagner (2017) [21] | ✓ | | ✓ | ✓ |
| Chen et al. (2017a) [22] | | ✓ | ✓ | ✓ |
| Chen et al. (2017b) [23] | | | ✓ | |
| Das et al. (2017) [24] | | | ✓ | ✓ |
| Gu et al. (2017) [25] | ✓ | | ✓ | |
| Jang et al. (2017) [26] | | | | ✓ |
| Moosavi et al. (2017) [27] | | | ✓ | ✓ |
| Xu et al. (2017) [28] | ✓ | | ✓ | ✓ |
| Kurakin et al. (2016) [29] | | | | ✓ |
| Moosavi et al. (2016) [30] | | | ✓ | ✓ |
| Papernot et al. (2016a) [31] | ✓ | | | |
| Papernot et al. (2016b) [32] | ✓ | | | |
| Goodfellow et al. (2014) [33] | | | | ✓ |

where $F()_C$ is the soft-label for the correct class, $F()_T$ is the soft-label for the target class, $p \in \{0, 1, 2, \infty\}$ is the constraint norm on $\epsilon_x$ and *th* is the threshold value for the constraint norm.

The constraint in the optimisation problem has the objective of disallowing perturbations that could make $x$ unrecognisable or change its correct class. Therefore, the constraint is itself a mathematical definition of what constitutes an imperceptible perturbation. Many different norms are used in the literature ($L_0$, $L_1$, $L_2$ and $L_\infty$). Intuitively, these norms allow for different types of attacks and thus different variations of defences. Table 2 shows the norm(s) used in assessment in existing literature.

For simplicity, we are narrowing the scope of this article to the image classification problem alone. However, the proposed attacks and the robustness assessment can be extended to other domain problems by modifying the constraint of the adversarial attacks according to the domain problem.

## Guaranteeing the quality of adversarial samples

Constraining the perturbation is decisive in adversarial samples to avoid; a) producing samples that can not be recognised by human beings, or b) producing samples that have changed their correct class by the amount of perturbation. However, restraining the total amount of perturbation is not enough as a small amount of perturbation concentrated in a few pixels might be able to create false adversarial samples. Therefore, a spatial constraint over the perturbation of pixels $P$ would be a desirable feature. This can be achieved mathematically as follows;

Given an image $x$ and its perturbed counterpart $x'$, it is possible to calculate $L_1$ norm between both matrices:

$$\|x - x'\|_1 \tag{6}$$

Constraining $L_1$ to be less than a certain number does not guarantee any spatial distribution constraint.

Let us define a set based on all non-zero pixel perturbations as follows:

$$N_z = \{P_i : \|P_i - P'_i\|_1 > 0\} \tag{7}$$

where $P_i$ and $P'_i$ are pixels from respectively the original image $x$ and the perturbed image $x'$ and $i$ is the image index. Both $N_z$ and its cardinality $|N_z|$ have information about perturbations' spatial distribution, and constraining any of these values would result in a spatially limited perturbation. Provided that $th$ is low enough, a modification preserving the white noise of that intensity would bound $|N_z| < th$.

Moreover, $|N_z|$ is precise $L_0$, demonstrating that $L_0$ is based on the set $N_z$, which stores spatial information about the differences. At the same time, the $L_1$ norm does not have this information.

Similarly, the $L_\infty$ norm can be rewritten as the following optimisation constraint:

$$\forall P_i \in x, \|P_i - P'_i\|_\infty \leq th \tag{8}$$

Notice that this constraint is also defined over the spatial distribution of perturbations.

Fig 1 gives empirical evidence of a misleading adversarial sample of an image that is constrained by $L_2 \leq 765$. Notice that this value is precisely the maximum change of one pixel, i.e., the maximum possible perturbation of the One-Pixel attack ($L_0 \leq 1$), which, when no limits are imposed over its spatial distribution, may create false adversarial samples. The reasoning behind the $L_0$ and $L_\infty$ are as follows, without altering much of the original sample, attacks can perturb a few pixels strongly ($L_0$), all pixels slightly ($L_\infty$) or a mix of both ($L_1$ and $L_2$). The hurdle is that $L_1$ and $L_2$, which mix both strategies vary enormously with the size of images, if not used with caution, may cause unrecognisable adversarial samples (Problem P3). Also, it is difficult to compare between methods using $L_1$ and $L_2$ norm because the amount of perturbations will often differ (Problem P4).

## Threshold attack (L$_\infty$ black-box attack)

The threshold attack optimizes the constrained optimization problem with the constraint $\|\epsilon_x\|_\infty \leq th$, i.e., it uses the $L_\infty$ norm. The algorithm searches in $\mathbb{R}^k$ space same as the input space. This is because the variables can be any input variation as long as the threshold is respected. In image classification problem $k = m \times n \times c$ where $m \times n$ is the size, and $c$ is the number of channels of the image.

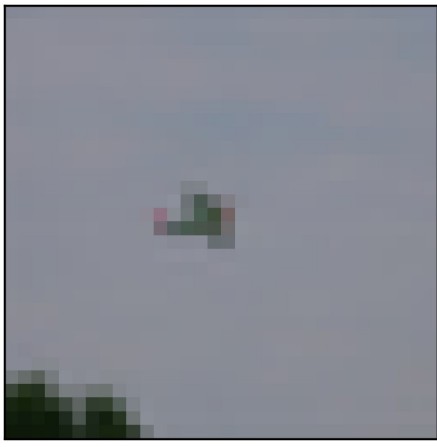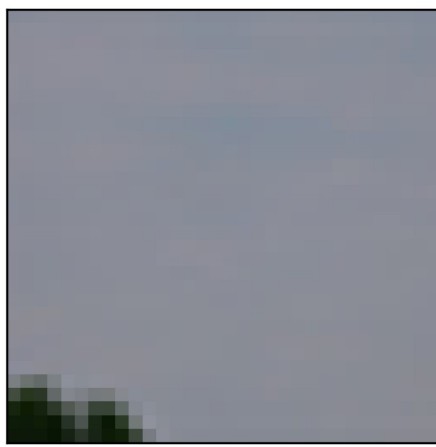

**Fig 1. Example of false adversarial sample.** We present an example of a false adversarial sample (right) and its respective original sample (left). The false adversarial sample is built with few total perturbations (i.e., low $L_1$ and $L_2$) but an unrecognisable final image (false adversarial sample). This results from the non-constrained spatial distribution of perturbations which is prevented if low $L_0$ or $L_\infty$ is used. This hypothetical attack has a $L_2$ of merely 356, well below the maximum $L_2$ for the One-Pixel ($L_0 \leq 1$) Attack (765).

## Few-Pixel attack ($L_0$ black-box attack)

The few-pixel attack is a variation of previously existing One-Pixel Attack [37]. It optimizes the constrained optimization problem by using the constraint $\|\epsilon_x\|_0 \leq th$, i.e., it uses the $L_0$ norm. The search variable is a combination of pixel values (depending on channels $c$ in the image) and position (2 values X, Y) for all of the pixels ($th$ pixels). Therefore, the search space is smaller than the threshold attack defined below with dimensions of $\mathbb{R}^{(2+c)\times th}$.

## Robustness levels

We propose robustness levels as machine learning algorithms might perform differently to varying amounts of perturbations. Robustness levels evaluate classifiers in a couple of $th$ thresholds. Explicitly, we define four levels of robustness 1, 3, 5, 10 for both of our $L_0$ Norm Attack and $L_\infty$ Norm Attack. We then name them respectively pixel and threshold robustness levels. Algorithms that pass a level of robustness (0% adversarial accuracy) are called level-threshold-safe or level-pixel-safe. For example, an algorithm that passes the level-one in threshold ($L_\infty$) attack is called 1-threshold-safe.

## Limitations

This proposed method takes into account adversarial attacks based on distance-based perturbations for generating imperceptible perturbations. However, there are other types of adversarial attacks like translation and rotation attacks [55] that do not change individual pixel values but instead change the image's orientation.

## Experimental results and discussions

In this section, we aim to validate the robustness assessment (Code is available at http://bit.ly/DualQualityAssessment) empirically as well as analyse the current state-of-the-art neural networks in terms of robustness.

## Preliminary tests

Tests on two neural networks are presented (ResNet [56] and CapsNet [57]). These tests are done to choose the black-box optimisation algorithm to be used for the further sections. The performance of both Differential Evolution (DE) [58] and Covariance Matrix Adaptation Evolution Strategy (CMA-ES) [59] are evaluated.

## Evaluating learning and defense systems

Preliminary Tests are extended to the seven different neural network architectures- WideResNet [60], DenseNet [61], ResNet [56], Network in Network (NIN) [62], All Convolutional Network (AllConv) [63], CapsNet [57], and LeNet [64]. We also evaluate three adversarial defences applied to the standard ResNet architecture- Adversarial training (AT) [16], Total Variance Minimization (TVM) [15], and Feature Squeezing (FS) [28]. We have chosen these defences as they are based on entirely different principles. In this way, the results achieved here can be extended to other similar types of defences in the literature.

## Analysing effect of threshold th on learning systems

We analyse the complete behaviour of our black-box attacks' adversarial accuracy without restricting the threshold's *th* value (robustness levels). Using this analysis, we prove the results using a fixed *th* in robustness levels is a reasonable approximation for our proposed robustness assessment.

## Evaluating other adversarial attacks

The evaluated learning systems are tested against other existing adversarial attacks such as- Fast Gradient Method (FGM) [33], Basic Iterative Method (BIM) [29], Projected Gradient Descent Method (PGD) [16], DeepFool [30], and NewtonFool [26]. This analysis further helps to demonstrate the necessity of duality in robustness assessment.

## Dependency of proposed adversarial attacks on classes

Here, we assess the dependence of proposed adversarial attacks on specific classes. The distribution analysis identifies and validates whether some classes are more challenging to attack than others.

## Transferability of adversarial samples

In this section, we apply and evaluate the transferability principle of adversarial samples. We verify the possibility of a speedy version of the proposed robustness assessment. We implement this by using already crafted adversarial samples to fool neural networks instead of a full-fledged optimisation. This would enable attacks to be significantly faster with a $O(1)$ time complexity.

## Adversarial sample distribution of robustness assessment

Here, we assess the dual-attack distribution of the Few-Pixel Attack and Threshold Attack. The analysis of this distribution demonstrates the necessity of such duality. The distribution of successful attacks is shown and analysed in this perspective.

**Table 3. Description of various parameters of different adversarial attacks.**

| Attack | | Parameters |
|---|---|---|
| FGM | | norm = $L_\infty$, $\epsilon = 8$, $\epsilon_{step} = 2$ |
| BIM | | norm = $L_\infty$, $\epsilon = 8$, $\epsilon_{step} = 2$, iterations = 10 |
| PGD | | norm = $L_\infty$, $\epsilon = 8$, $\epsilon_{step} = 2$, iterations = 20 |
| DeepFool | | iterations = 100, $\epsilon = 0.000001$ |
| NewtonFool | | iterations = 100, eta = 0.01 |
| $L_0$ Attack | Common | Parameter Size = 5, |
| | DE | NP = 400, Number of Generations = 100, CR = 1 |
| | CMA-ES | Function Evaluations = 40000, $\sigma = 31.75$ |
| $L_\infty$ Attack | Common | Parameter Size = 3072, |
| | DE | NP = 3072, Number of Generations = 100, CR = 1 |
| | CMA-ES | Function Evaluations = 39200, $\sigma = th/4$ |

## Experimental settings

We use the CIFAR-10 dataset [65] to evaluate our robustness assessment. Table 3 gives the parameter description of various adversarial attacks used. All the pre-existing adversarial attacks used in the article have been evaluated using Adversarial Robustness 360 Toolbox (ART v1.2.0) [66].

For our $L_0$ and $L_\infty$ Attacks, we use the canonical versions of the DE and CMA-ES algorithms to have a clear standard. DE uses a repair method in which values beyond range are set to random points within the valid range. While in CMA-ES, to satisfy the constraints, a simple repair method is employed in which pixels that surpass the minimum/maximum are brought back to the minimum/maximum value. Moreover, a clipping function is used to keep values inside the feasible region.

The constraint is always satisfied because the number of parameters is itself modelled after the constraint. In other words, when searching for one pixel perturbation, the number of variables are fixed to pixel values (three values) plus position values (two values). Therefore it will always modify only one pixel, respecting the constraint. To force the pixel values to be within range, a simple clipping function is used for pixel values since the optimisation is done in real values, For position values, a modulo operation is executed.

## Preliminary tests: Choosing the optimization algorithm

Table 4 shows the adversarial accuracy results performed over 100 random samples. Here adversarial accuracy corresponds to the accuracy of the adversarial attack to create adversarial samples which fool neural networks. In other words, the success rate of the adversarial attack. Both proposed attacks can craft adversarial samples in all levels of robustness. This demonstrates that black-box attacks can still reach more than 80% adversarial accuracy in state-of-the-art neural networks without knowing anything about the learning system and in a constrained setting.

Concerning the comparison of CMA-ES and DE, the outcomes favour the choice of CMA-ES for the robustness assessment. Both CMA-ES and DE perform likewise for the Few-Pixel Attack, with both DE and CMA-ES having the same number of wins. However, for the Threshold Attack, the performance varies significantly. CMA-ES, this time always wins (eight wins) against DE (no win).

This domination of CMA-ES is expected since the Threshold ($L_\infty$) Attack has a high dimensional search space which is more suitable for CMA-ES. This happens partly because

**Table 4. Adversarial accuracy results for Few-Pixel ($L_0$) and Threshold ($L_\infty$) attacks with DE and CMA-ES.**

| Model | Attack Optimiser | Adversarial Accuracy | | | |
|---|---|---|---|---|---|
| | | th = 1 | th = 3 | th = 5 | th = 10 |
| Few-Pixel ($L_0$) Attack | | | | | |
| ResNet | DE | **24%** | **70%** | **75%** | 79% |
| | CMA-ES | 12% | 52% | 73% | **85%** |
| CapsNet | DE | **21%** | 37% | **49%** | **57%** |
| | CMA-ES | 20% | **39%** | 40% | 41% |
| Threshold ($L_\infty$) Attack | | | | | |
| ResNet | DE | 5% | 23% | 53% | 82% |
| | CMA-ES | **33%** | **71%** | **76%** | **83%** |
| CapsNet | DE | 11% | 13% | 15% | 23% |
| | CMA-ES | **13%** | **34%** | **72%** | **97%** |

DE's operators may allow some variables to converge prematurely. CMA-ES, on the other hand, is consistently generating slightly different solutions while evolving a distribution.

In these preliminary tests, CapsNet was shown overall superior to ResNet. Few-pixel ($L_0$) Attack reach 85% adversarial accuracy for ResNet when ten pixels are modified. CapsNet, on the other hand, is more robust to Few-Pixel Attacks, allowing them to reach only 52% and 41% adversarial accuracy when ten pixels are modified for DE and CMA-ES, respectively. CapsNet is less robust than ResNet to the Threshold Attack with $th = 10$ in which almost all images were vulnerable (97%).

At the same time, CapsNet is reasonably robust to 1-threshold-safe (only 13%). ResNet is almost equally not robust throughout, with low robustness even when $th = 3$, losing to CapsNet in robustness in all other values of $th$ of the threshold attack. These preliminary tests also show that different networks have different robustness. This is not only regarding the type of attacks ($L_0$ and $L_\infty$) and the degree of attack (e.g., 1-threshold and 10-threshold attacks have very different results on CapsNet).

## Evaluating learning and defense systems

Table 5 extends the CMA-ES based Few-Pixel ($L_0$) and Threshold ($L_\infty$) attacks on various neural network architectures: WideResNet [60], DenseNet [61], ResNet [56], Network in Network (NIN) [62], All Convolutional Network (AllConv) [63], CapsNet [57], and LeNet [64]. We also evaluate with three contemporary defences: Adversarial training (AT) [16], Total Variance Minimization (TVM) [15], and Feature Squeezing (FS) [28].

Results in bold (Only for learning systems and not defensive systems) are the lowest adversarial accuracy and other results within 5% margin from the lowest one. For CapsNet only 88 samples could be attacked with maximum $th = 127$ for $L_0$ Attack. Twelve samples could not be overwhelmed when the $th < 128$.

Here, considering an existing variance of results, we consider results within 5% of the lowest to be equally good. Considering the number of bold results for each neural network, a qualitative measure of robustness CapsNet and AllConv can be considered the most robust with five bold results. The third place in robustness achieves only three bold results and is far away from the prime performers.

Regarding the adversarial training, it is easier to attack with the Few-Pixel Attack than with the Threshold Attack. This result should derive from the fact that the adversarial samples used in adversarial training contained images from Projected Gradient Descent (PGD) Attack,

**Table 5. Adversarial accuracy results for $L_0$ and $L_\infty$ attacks over 100 random samples.**

| Model and Standard Accuracy | | Adversarial Accuracy | | | |
|---|---|---|---|---|---|
| | | th = 1 | th = 3 | th = 5 | th = 10 |
| *Few-Pixel ($L_0$) Attack* | | | | | |
| WideResNet | 95.12% | **11%** | 55% | 75% | 94% |
| DenseNet | 94.54% | **9%** | 43% | 66% | 78% |
| ResNet | 92.67% | **12%** | 52% | 73% | 85% |
| NIN | 90.87% | 18% | 62% | 81% | 90% |
| AllConv | 88.46% | **11%** | **31%** | 57% | 77% |
| CapsNet | 79.03% | 21% | 37% | **49%** | **57%** |
| LeNet | 73.57% | 58% | 86% | 94% | 99% |
| AT | 87.11% | 22% | 52% | 66% | 86% |
| TVM | 47.55% | 16% | 12% | 20% | 24% |
| FS | 92.37% | 17% | 49% | 69% | 78% |
| *Threshold ($L_\infty$) Attack* | | | | | |
| WideResNet | 95.12% | 15% | 97% | 98% | 100% |
| DenseNet | 94.54% | 23% | 68% | **72%** | **74%** |
| ResNet | 92.67% | 33% | 71% | **76%** | 83% |
| NIN | 90.87% | **11%** | 86% | 88% | 92% |
| AllConv | 88.46% | **9%** | 70% | **73%** | **75%** |
| CapsNet | 79.03% | **13%** | **34%** | 72% | 97% |
| LeNet | 73.57% | 44% | 96% | 100% | 100% |
| AT | 87.11% | 3% | 12% | 25% | 57% |
| TVM | 47.55% | 4% | 4% | 6% | 14% |
| FS | 92.37% | 26% | 63% | 66% | 74% |

which is $L_\infty$ norm-based attack. Therefore, it suggests that *given an attack bias that differs from the invariance bias used to train the networks, the attack can easily succeed.*

Regarding TVM, the attacks were less successful. We trained a ResNet on TVM modified images, and, albeit in many trials with different hyper-parameters, we were able to craft a classifier with at best 47.55% accuracy. This is a steep drop from the 92.37% accuracy of the original ResNet and happens because TVM was initially conceived for Imagenet and did not scale well to CIFAR-10. However, as the original accuracy of the model trained with TVM is also not high; therefore, even with a small attack percentage of 24%, the resulting model accuracy is 35%.

Attacks on Feature Squeezing had relatively high adversarial accuracy in both $L_0$ and $L_\infty$ attacks. Moreover, both types of attacks had similar accuracy, revealing a lack of bias in the defence system.

Notice that none of the neural networks was able to reduce low *th* attacks to 0%. This illustrates that although robustness may differ between neural networks, none of them can effectively overcome even the lowest level of perturbation feasible.

Moreover, since a *th* = 5 is enough to achieve around 70% accuracy in many settings, this suggests that achieving 100% adversarial accuracy may depend more on a few samples which are harder to attack, such as samples far away from the decision boundary. Consequently, the focus on 100% adversarial accuracy rather than the amount of threshold might give preference to methods that set a couple of input projections far away from others without improving the accuracy overall. An example can be examined by making some input projections far away enough to make them harder to attack.

The difference in the behaviour of $L_0$ and $L_\infty$ Norm Attacks shows that the robustness is achieved with some trade-offs. This further justifies the importance of using both metrics to evaluate neural networks.

## Analysing effect of threshold *th* on learning systems

We plot here the adversarial accuracy with the increase of *th* (Figs 2 and 3) to evaluate how networks behave with the increase in the threshold. These plots reveal an even more evident behaviour difference for the same method when attacked with either $L_0$ or $L_\infty$ norm of attacks. It shows that the curve inclination itself is different. Therefore, $L_0$ and $L_\infty$ Attacks scale differently.

From Figs 2 and 3, two classes of curves can be seen. CapsNet behaves in a class of its own while the other networks behave similarly. CapsNet, which has an entirely different architecture with dynamic routing, shows that a very different robustness behaviour is achieved. LeNet is justifiably lower because of its lower accuracy and complexity.

To assess the quality of the algorithms about their curves, the Area Under the Curve (AUC) is calculated by the trapezoidal rule. defined as:

$$\text{AUC} = \Delta n_a \left( \frac{th_1}{2} + th_2 + th_3 + \ldots + th_{n-1} + \frac{th_n}{2} \right) \tag{9}$$

where $n_a$ is the number of images attacked and $th_1, th_2, \ldots th_n$ are different values of *th* threshold for a maximum of $n = 127$. Table 6 shows a quantitative evaluation of Fig 2 by calculating AUC.

There is no network that is robust in both attacks. CapsNet is the most robust neural network for $L_0$ attacks, while AllConv wins while being followed closely by other neural networks for $L_\infty$. Although requiring many more resources to be drawn, the curves here result in the same conclusion achieved by Table 5. Therefore, the previous results are a good approximation of the behaviour promptly.

## Evaluating other adversarial attacks

We evaluated our assessed neural networks further against well-known adversarial attacks such as Fast Gradient Method (FGM) [33], Basic Iterative Method (BIM) [29], Projected Gradient Descent Method (PGD) [16], DeepFool [30], and NewtonFool [26]. Please, note that for FGM, BIM, PGD attacks $\epsilon = 8$(Default Value) $\approx th = 10$ of $L_\infty$ Attack on our robustness scales. While DeepFool and NewtonFool do not explicitly control the robustness scale.

Table 7 compares the existing attacks with our proposed attacks. Notice that, although all the existing attacks can fool neural networks. We notice some peculiar results, like DeepFool Attack, which was less successful against the LeNet, which was most vulnerable to our proposed attacks (Table 5). Moreover, ResNet and DenseNet had much better robustness for the existing attacks than our attacks.

Note that the objective of this article is not to propose better or more effective attacking methods but rather to propose an assessment methodology and its related duality conjecture (the necessity of evaluating both $L_0$ and $L_\infty$ Attacks). However, the proposed Threshold $L_\infty$ Attack in the assessment methodology is more accurate than other attacks while requiring less amount of perturbation. The Threshold Attack requires more minor perturbation than the One-Pixel attack (only circa 12% of the amount of perturbation of the One-Pixel Attack $th = 1$), which was already considered one of the most extreme attacks. This sets up an even lower threshold to perturbation, which is inevitable to fool neural networks.

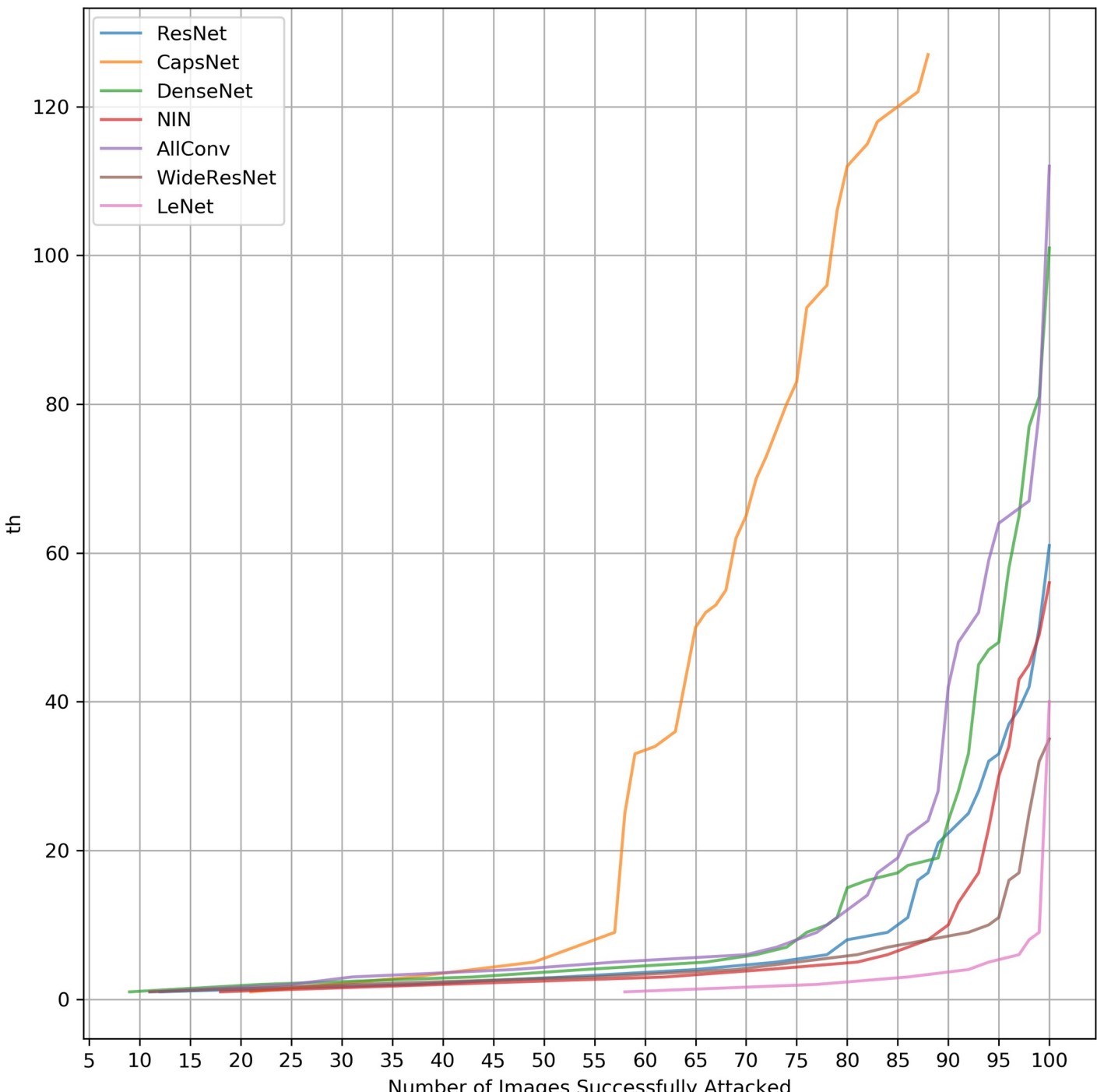

**Fig 2. Adversarial accuracy per *th* for $L_0$ attack.**

Notice that the existing attacks' behaviour is similar to our $L_\infty$ Attack (Table 7), which suggests that the current evaluations of the neural networks focus on increasing the robustness based on $L_\infty$ norm. However, our study shows that the behaviour of $L_0$ norm differs from the $L_\infty$ norm (Table 7), and the robustness for the $L_\infty$ Norm may not be sufficient to study the robustness of the neural networks.

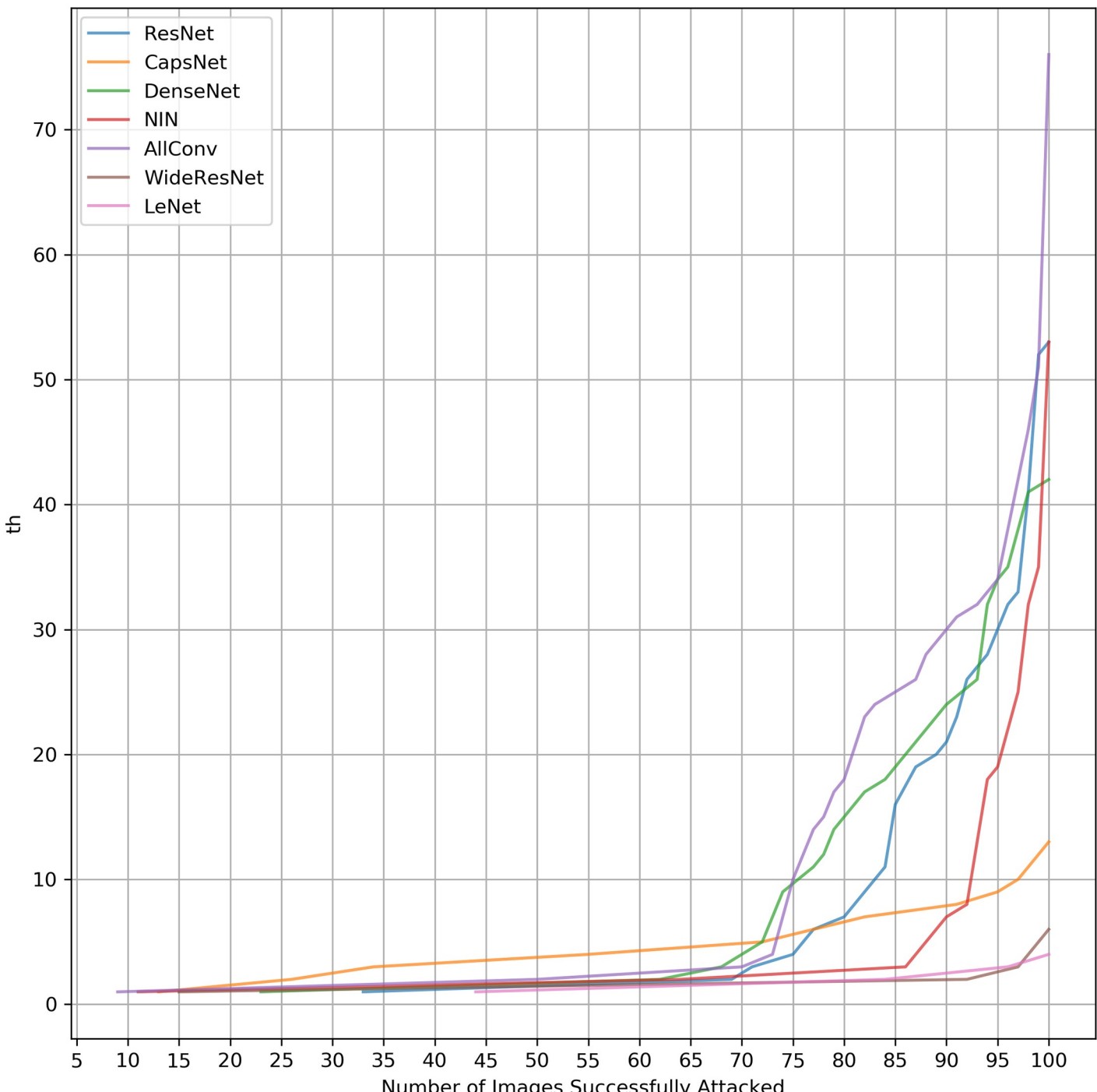

**Fig 3. Adversarial accuracy per *th* for $L_\infty$ attack.**

## Dependency of proposed adversarial attacks on classes

We further separated the adversarial accuracy (Table 5) into classes (Fig 4). This is to evaluate the dependence of proposed adversarial attacks on specific classes.

**Table 6. Area under the curve (AUC) for both Few-Pixel ($L_0$) and Threshold ($L_\infty$) black-box attacks.**

| Model | $L_0$ Attack | $L_\infty$ Attack |
|---|---|---|
| WideResNet | 425.0 | 141.5 |
| DenseNet | 989.5 | 696.0 |
| ResNet | 674.0 | 575.5 |
| NIN | 528.0 | 364.0 |
| AllConv | 1123.5 | **849.0** |
| CapsNet | **2493.0** | 404.5 |
| LeNet | 137.5 | 104.0 |

Fig 4 shows an already known feature that some classes are more natural to attack than others. For example, the columns for bird and cat classes are visually darker than the frog and truck classes for all diagrams. This happens because classes with similar features and, therefore, closer decision boundaries are more natural to attack.

Interestingly, Fig 4 reveals that neural networks tend to be harder to attack in only a few classes. This may suggest that these networks encode some classes far away from others (e.g., projection of the features of these classes into a different vector). Consequently, the reason for their relative robustness may lie in a simple construction of the decision boundary with a few distinct and sharply separated classes.

## Transferability of adversarial samples

Fig 5 assess different neural network architectures qualitatively based on the transferability of adversarial samples. If adversarial samples from one model can be used to attack different models and defences, it would be possible to create an ultra-fast robustness assessment. Generally speaking, transferability is a quick assessment method that, when used with many different types of adversarial samples, approximates the model's robustness.

**Table 7. Adversarial accuracy of the proposed $L_0$ and $L_\infty$ black-box attacks used in the robustness assessment compared with other methods from the literature.**

| Adversarial Attacks | | WideResNet | DenseNet | ResNet | NIN | AllConv | CapsNet | LeNet |
|---|---|---|---|---|---|---|---|---|
| FGM | | 69% (159.88) | 50% (120.03) | 52% (124.70) | 72% (140.46) | 67% (155.95) | 70% (208.89) | 84% (152.37) |
| BIM | | 89% (208.44) | 52% (160.34) | 55% (164.64) | 74% (216.97) | 69% (273.90) | 82% (361.63) | 89% (345.27) |
| PGD | | 89% (208.49) | 52% (160.38) | 55% (164.64) | 74% (216.96) | 69% (274.15) | 84% (370.90) | 89% (357.34) |
| DeepFool | | 60% (613.14) | 60% (478.03) | 58% (458.57) | 59% (492.90) | 51% (487.46) | 87% (258.08) | 31% (132.32) |
| NewtonFool | | 82% (63.13) | 50% (53.89) | 54% (51.56) | 66% (54.78) | 61% (61.05) | 90% (1680.83) | 84% (49.61) |
| Few-Pixel ($L_0$) Attack | $th = 1$ | 20% (181.43) | 20% (179.48) | 29% (191.73) | 28% (185.09) | 24% (172.01) | 29% (177.86) | 61% (191.69) |
| | $th = 3$ | 54% (276.47) | 50% (270.50) | 63% (275.57) | 62% (274.91) | 49% (262.66) | 43% (247.97) | 89% (248.21) |
| | $th = 5$ | 75% (326.14) | 68% (315.53) | 79% (314.27) | 81% (318.71) | 67% (318.99) | 52% (300.19) | 96% (265.18) |
| | $th = 10$ | 91% (366.60) | 81% (354.42) | 90% (342.56) | 93% (354.61) | 81% (365.10) | 63% (359.55) | 98% (271.90) |
| Threshold ($L_\infty$) Attack | $th = 1$ | 30% (39.24) | 38% (39.24) | 43% (39.27) | 23% (39.23) | 23% (39.21) | 13% (39.09) | 47% (39.28) |
| | $th = 3$ | 92% (65.07) | 69% (53.89) | 74% (52.82) | 81% (72.29) | 72% (68.11) | 34% (70.79) | 96% (62.86) |
| | $th = 5$ | 95% (67.84) | 72% (56.81) | 77% (55.38) | 85% (77.09) | 76% (72.45) | 72% (130.80) | 99% (66.42) |
| | $th = 10$ | 98% (70.70) | 78% (67.63) | 83% (64.50) | 90% (84.20) | 79% (77.76) | 97% (184.93) | 100% (66.65) |

The value in the brackets represents the Mean $L_2$ score of the adversarial sample with the original sample. *The results were drawn by attacking a different set of samples from previous tests. Therefore the accuracy results may differ slightly from previous tables.*

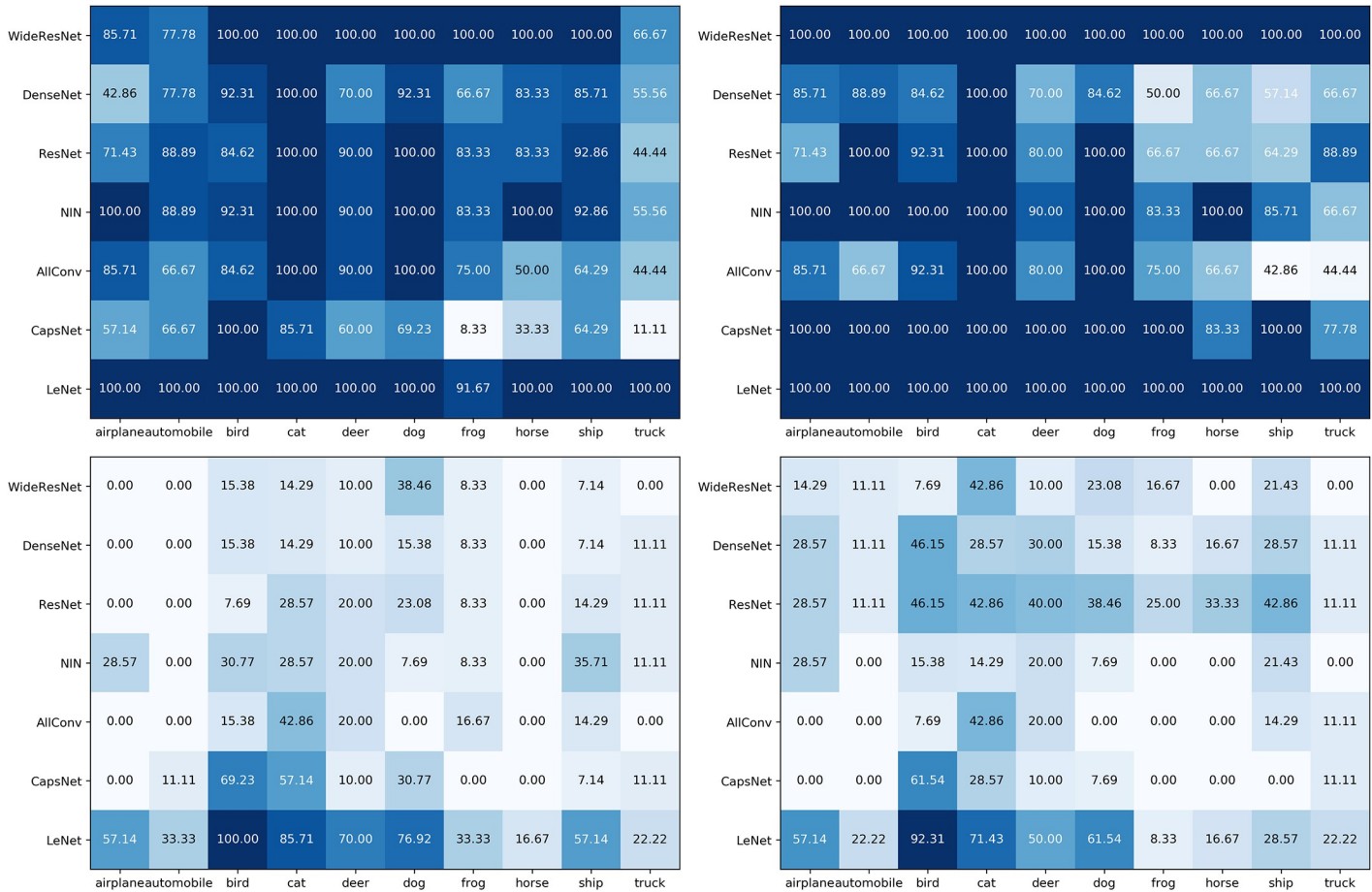

**Fig 4. Adversarial accuracy from Table 5 across classes.** The two diagrams at left and right are respectively $L_0$ and $L_\infty$ attacks. The top diagrams used $th = 10$ while the bottom ones used $th = 1$.

Beyond being a faster method, the transferability of samples can ignore any masking of gradients, which makes it hard to search but not to transfer. This shows that the vulnerability in neural networks is still there but hidden.

This approximation is different, not better, or worse from the usual analysis because, (a) it is not affected by how difficult it is to search adversarial samples, taking into account only their existence, and (b) it measures the accuracy of commonly found adversarial samples rather than all searchable ones.

Therefore, in low *th* values, transferability can be used as a qualitative measure of robustness. However, its values are not equivalent to or close to real adversarial accuracy. Thus, it serves only as a lower bound.

Interestingly, the transferability is primarily independent of the type of attack ($L_0$ or $L_\infty$), with most of the previously discussed differences disappearing. Some differences, like $L_0$ attacks, are less accurate than most of the $L_\infty$ ones. This suggests that positions of pixel and their variance are relatively more model-specific than small changes in the whole image.

## Adversarial sample distribution of robustness assessment

We analyse the distribution of our proposed attacks across samples to understand the importance of the duality for the proposed robustness assessment. In some cases, the distribution of

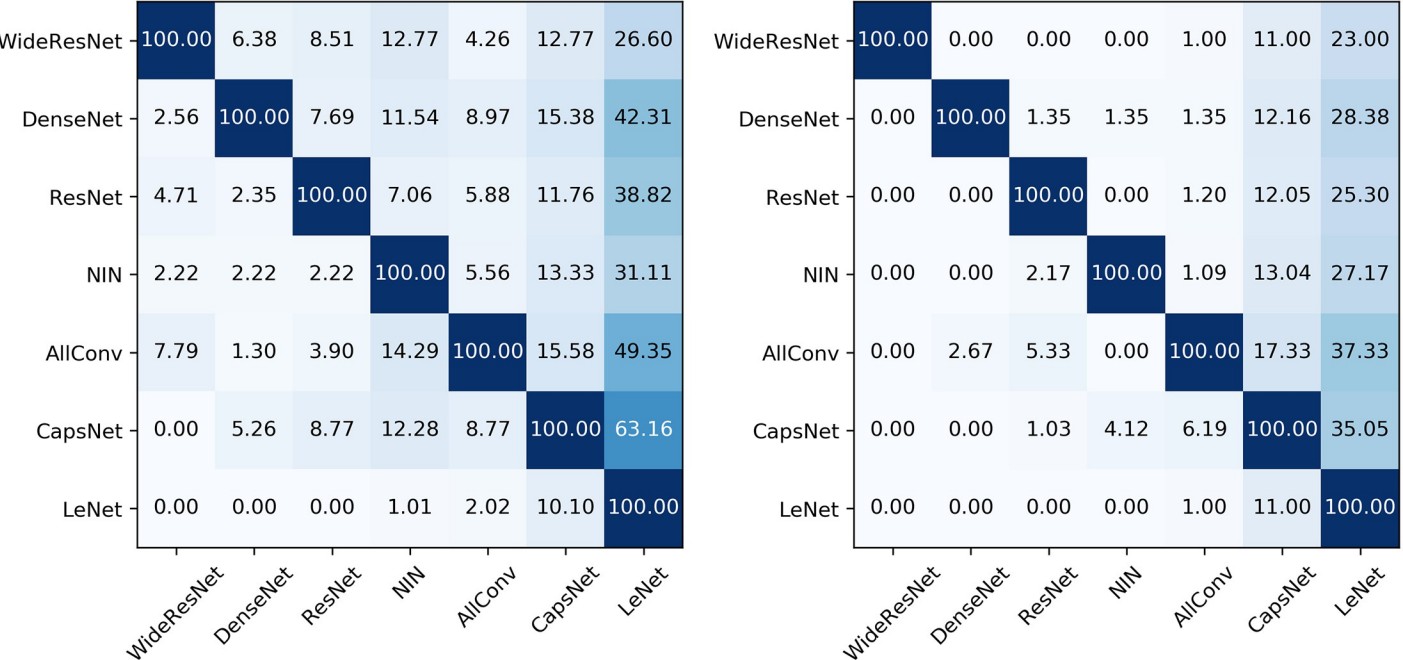

**Fig 5. Transferability of adversarial samples.** Accuracy of adversarial samples when transferring from the a given source model (row) to a target model (column) for both $L_\infty$ black-box Attacks (left) and $L_0$ black-box Attacks (right). The source of the adversarial samples is on the y-axis with the target model on the x-axis. The adversarial samples were acquired from 100 original images attacked with *th* varying mostly from one to ten. The maximum value of *th* is set to 127.

samples for $L_0$ and $L_\infty$ can be easily verified by the difference in adversarial accuracy. For example, CapsNet is more susceptible to $L_\infty$ than $L_0$ types of attacks while for adversarial training [16] the opposite is true (Table 5). Naturally, adversarial training depends strongly on the adversarial samples used in training, Therefore, different robustness could be acquired depending on the type of adversarial samples used.

Moreover, the distribution shows that even when adversarial accuracy seems close, the distribution of $L_0$ and $L_\infty$ Attacks may differ. For example, the adversarial accuracy on ResNet for both $L_0$ and $L_\infty$ with *th* = 10 differ by mere 2%. However, the distribution of adversarial samples shows that around 17% of the samples can only be attacked by either one attack type (Fig 6). Thus, evaluating both $L_0$ and $L_\infty$ are essential to verify the robustness of a given neural network or adversarial defence. Moreover, this is true even when a similar adversarial accuracy is observed.

## Conclusions

This article proposes a model agnostic adversarial robustness assessment for adversarial machine learning, especially for neural networks. By investigating the various neural network architectures as well as arguably the contemporary adversarial defences, it was possible to:

(a). show that robustness to $L_0$ and $L_\infty$ Norm Attacks differ significantly, which is why the duality should be taken into consideration,

(b). verify that current methods and defences, in general, are vulnerable even for $L_0$ and $L_\infty$ black-box Attacks of low threshold *th*, and,

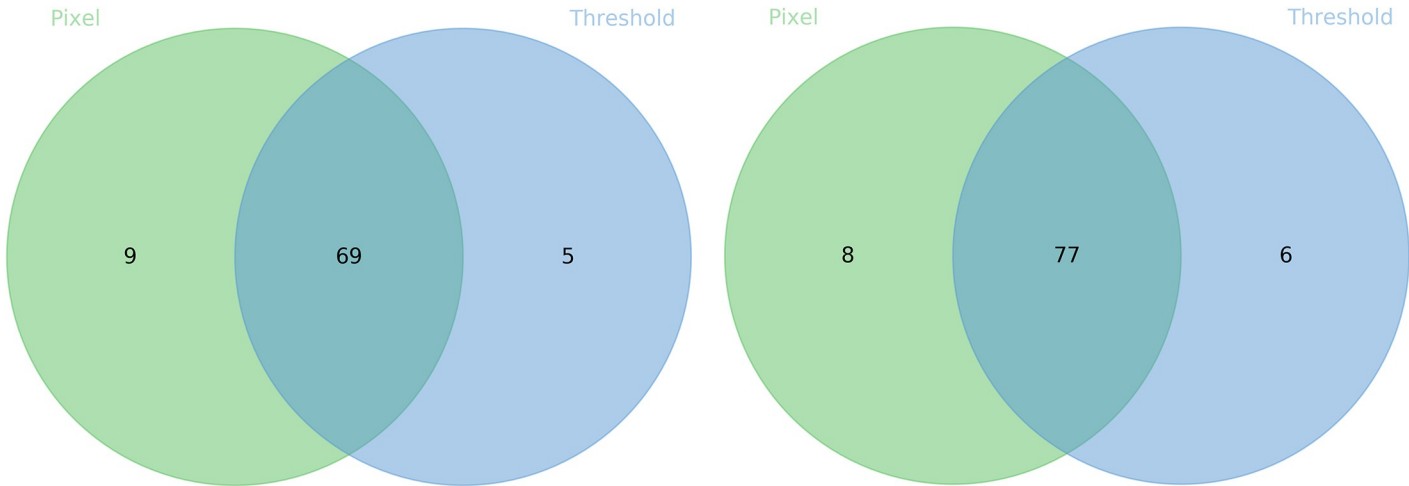

**Fig 6. Distribution of adversarial samples.** Distribution of adversarial samples found on DenseNet (left) and ResNet (right) using $th = 10$ with both few-pixel ($L_0$) and threshold ($L_\infty$) Attacks.

 (c). validate the robustness assessment with robustness level as a fair and efficient approximation to the full accuracy per threshold curve.

Interestingly, the evaluation of the proposed $L_\infty$ black-box Attack based on CMA-ES required only circa 12% of the amount of perturbation used by the One-Pixel Attack while achieving similar accuracy. Moreover, the evaluation of adversarial attacks suggests that current adversarial defences defend deep neural networks from a single type of adversarial attack but fails to defend against different types of adversarial attack. Therefore, adversarial defences successful against an adversarial attack might be vulnerable to the same adversarial attack with different constraints.

Thus, this article analyses the robustness of neural networks and defences by elucidating the problems and proposing solutions to them. Hopefully, the proposed robustness assessment and analysis on current neural networks' robustness will help develop more robust neural networks and hybrids alike. This opens up the opportunity to study robustness of neural networks from a broad perspective. As a future work, this assessment can be extended to non-distance based adversarial attacks like translation and rotation attacks [55], and colour channel perturbation attack [67] to make robustness assessment more general.

## Acknowledgments

We would like to thank Prof. Junichi Murata for his kind support without which it would not be possible to conduct this research.

## Author Contributions

**Conceptualization:** Shashank Kotyan, Danilo Vasconcellos Vargas.

**Data curation:** Shashank Kotyan.

**Formal analysis:** Shashank Kotyan.

**Funding acquisition:** Danilo Vasconcellos Vargas.

**Investigation:** Shashank Kotyan.

**Methodology:** Shashank Kotyan.

**Project administration:** Danilo Vasconcellos Vargas.

**Resources:** Danilo Vasconcellos Vargas.

**Software:** Danilo Vasconcellos Vargas.

**Supervision:** Danilo Vasconcellos Vargas.

**Validation:** Shashank Kotyan.

**Visualization:** Shashank Kotyan.

**Writing – original draft:** Shashank Kotyan.

**Writing – review & editing:** Shashank Kotyan, Danilo Vasconcellos Vargas.

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
