## [Decision Letter · Decision Letter 0]

2 Feb 2022

PONE-D-22-00930Adversarial Robustness Assessment Why in evaluation both L0 and L∞ attacks are necessaryPLOS ONE

Dear Dr. Kotyan,

Thank you for submitting your manuscript to PLOS ONE. After careful consideration, we feel that it has merit but does not fully meet PLOS ONE’s publication criteria as it currently stands. Therefore, we invite you to submit a revised version of the manuscript that addresses the points raised during the review process.

We look forward to receiving your revised manuscript.

Kind regards,

Sathishkumar V E

Academic Editor

PLOS ONE

Journal Requirements:

"This work was supported by JST, ACT-I Grant Number JP-50243 and JSPS KAKENHI Grant Number JP20241216. Additionally, we would like to thank Prof. Junichi Murata for the kind support without which it would not be possible to conduct this research."

"DV, JST-Act-I Grant Number JP-50243, https://www.jst.go.jp/kisoken/act-i/index.html

DV, JSPS KAKENHI Grant Number JP20241216, " ext-link-type="uri" xlink:type="simple">https://www.jsps.go.jp/english/e-grants/"

Reviewers' comments:

Reviewer's Responses to Questions

**Comments to the Author**

1. Is the manuscript technically sound, and do the data support the conclusions?

Reviewer #2: Yes

Reviewer #3: Yes

2. Has the statistical analysis been performed appropriately and rigorously? 

Reviewer #2: Yes

Reviewer #3: N/A

3. Have the authors made all data underlying the findings in their manuscript fully available?

Reviewer #2: Yes

Reviewer #3: Yes

4. Is the manuscript presented in an intelligible fashion and written in standard English?

Reviewer #2: No

Reviewer #3: Yes

5. Review Comments to the Author

Reviewer #2: This paper is generally well written and can be accepted for publication after need the following minor revisions:

1. The authors must carefully check the language and English grammar. There are also some typos.

2. Why the Authors have stated that the effectiveness of the L0 and L∞ attacks?. An additional explanation would be helpful for potential readers.

Reviewer #3: Review Report:

Title: Adversarial Robustness Assessment:

Why in evaluation both L_0 and L_{\\infty} attacks are necessary

The paper is devoted to analysis of evaluation both L_0 and L_{\\infty} attacks are necessary

However, the following points should be further improved:

1. Some sentence throughout the paper are strange which make the paper lack

readability. Author should check and correct it.

2. How to construct the derivation (2), need detailed explanations.

3. The main contributions of the paper should be clearly stated

4. Discuss the technical difficulty in dealing with proposed scheme.

5. One or two remarks should be given.

6. There are some typos and grammatical errors. The authors should check carefully and correct seriously.

7. 9. According to the topic of the paper, the authors should discuss some interesting problem as

future work in conclusion, such as 10.1109/ACCESS.2021.3123058; 10.1016/j.jfranklin.2020.01.016; 10.1109/ACCESS.2021.3060044.

8. There are too many alignment issues. Authors should correct it.

6. PLOS authors have the option to publish the peer review history of their article (what does this mean?). If published, this will include your full peer review and any attached files.

Reviewer #2: No

Reviewer #3: No

---

## [Author Response · Author response to Decision Letter 0]

28 Feb 2022

Reviewer #2: The authors must carefully check the language and English grammar. There are also some typos.

Response:The article is revised and has been corrected for typing and grammatical errors. 

Reviewer #2: Why the Authors have stated that the effectiveness of the L0 and L∞ attacks?. An additional explanation would be helpful for potential readers.

Response:Thank you for pointing it out. To improve the clarity on the topic, our abstract is revised to state explicitly that the effectiveness of L0 and L∞ attacks used in the article highlight the fact that neural networks are vulnerable to either one or both types of attacks, even when equipped with adversarial defences. Therefore, it is essential to evaluate both types of attacks.

Reviewer #3: Some sentence throughout the paper are strange which make the paper lack readability. Author should check and correct it. There are some typos and grammatical errors. The authors should check carefully and correct seriously. There are too many alignment issues. Authors should correct it.

Response:The readability of the article is improved with simpler sentence structures and also has been corrected for typing and grammatical errors. 

Reviewer #3: How to construct the derivation (2), need detailed explanations.

Response:We appreciate your feedback on pointing this out. To improve the clarity on the understanding of the derivation, further details have been updated in the text. The derivation of (2) follows from the definition of adversarial samples that, adversarial samples have a different classification than the original sample, thus effectively inducing misclassification. 

Reviewer #3: The main contributions of the paper should be clearly stated.

Response:In order to highlight and clearly state our contributions in this article according to the feedback, the paragraph title “Contributions” is added to the content where the contributions are discussed. 

Reviewer #3: Discuss the technical difficulty in dealing with proposed scheme.

Response:The technical difficulty and limitations of the proposed scheme are now discussed in the revised section “Guaranteeing the Quality of Adversarial Samples”.

Reviewer #3: One or two remarks should be given. According to the topic of the paper, the authors should discuss some interesting problem as future work in conclusion, such as 10.1109/ACCESS.2021.3123058; 10.1016/j.jfranklin.2020.01.016; 10.1109/ACCESS.2021.3060044.

Response:Our conclusions are revised to provide a few remarks and discuss interesting future research directions which could be pursued.

---

## [Decision Letter · Decision Letter 1]

8 Mar 2022

Adversarial Robustness Assessment Why in evaluation both L0 and L∞ attacks are necessary

PONE-D-22-00930R1

Dear Dr. Kotyan,

We’re pleased to inform you that your manuscript has been judged scientifically suitable for publication and will be formally accepted for publication once it meets all outstanding technical requirements.

Kind regards,

Sathishkumar V E

Academic Editor

PLOS ONE

Additional Editor Comments (optional):

Reviewers' comments:

Reviewer's Responses to Questions

**Comments to the Author**

1. If the authors have adequately addressed your comments raised in a previous round of review and you feel that this manuscript is now acceptable for publication, you may indicate that here to bypass the “Comments to the Author” section, enter your conflict of interest statement in the “Confidential to Editor” section, and submit your "Accept" recommendation.

Reviewer #2: (No Response)

Reviewer #3: (No Response)

2. Is the manuscript technically sound, and do the data support the conclusions?

Reviewer #2: (No Response)

Reviewer #3: (No Response)

3. Has the statistical analysis been performed appropriately and rigorously? 

Reviewer #2: (No Response)

Reviewer #3: (No Response)

4. Have the authors made all data underlying the findings in their manuscript fully available?

Reviewer #2: (No Response)

Reviewer #3: (No Response)

5. Is the manuscript presented in an intelligible fashion and written in standard English?

Reviewer #2: (No Response)

Reviewer #3: (No Response)

6. Review Comments to the Author

Reviewer #2: (No Response)

7. PLOS authors have the option to publish the peer review history of their article (what does this mean?). If published, this will include your full peer review and any attached files.

Reviewer #2: No

Reviewer #3: No

---

## [Editor Report · Acceptance letter]

6 Apr 2022

PONE-D-22-00930R1 

Adversarial Robustness Assessment:
Why in evaluation both *L*_0_ and *L*_∞_ attacks are necessary 

Dear Dr. Kotyan:

I'm pleased to inform you that your manuscript has been deemed suitable for publication in PLOS ONE. Congratulations! Your manuscript is now with our production department. 

Kind regards, 

on behalf of

Dr. Sathishkumar V E 

Academic Editor

PLOS ONE